# Integrated Crop-Nitrogen Management Improves Tomato Yield and Root Architecture and Minimizes Soil Residual N

Changqing Li [1,2], Yahao Li [1,3], Dongyu Cui [1], Yanmei Li [1], Guoyuan Zou [1], Rodney Thompson [4], Jiqing Wang [3,*] and Jungang Yang [1,*]

1 Institute of Plant Nutrition, Resources and Environment, Beijing Academy of Agriculture and Forestry Sciences, Beijing 100097, China; lixiaoqing19960815@163.com (C.L.); liyahao_1996@163.com (Y.L.); 18713022292@163.com (D.C.); liyanmei0101@163.com (Y.L.); gyzou@163.com (G.Z.)
2 College of Resources and Environmental Sciences, Hebei Agricultural University, Baoding 071001, China
3 College of Agriculture and Forestry Sciences, Hebei North University, Zhangjiakou 075000, China
4 Department of Agronomy, University of Almeria, 04120 La Cañada, Spain; rodney@ual.es
* Correspondence: wjq-72@126.com (J.W.); jungangyang@163.com (J.Y.)

**Abstract:** Sustainable intensification of protected vegetable crops entails increasing yield while reducing environmental impact and labor input. To explore a comprehensive strategy for high yielding, highly efficient and high quality production of greenhouse tomato (*Solanum lycopersicum* L.), an integrated crop-nitrogen management (ICNM) strategy was compared to farmers' traditional management (TM) in a field experiment in a solar greenhouse. A split-plot block design was used. The main factor was the management strategy of ICNM and TM. The secondary factor was four different basal fertilizer treatments, being a control (CK; 0 kg N ha$^{-1}$), carbon-based urea (BU; 100 kg N ha$^{-1}$), controlled release urea (CU; 100 kg N ha$^{-1}$), and conventional compound fertilizer (CF; 100 kg N ha$^{-1}$). An additional 200 kg N ha$^{-1}$ through drip irrigation as topdressing was used. Tomato fruit yield, N uptake, and N partial productivity with ICNM were significantly higher than with TM, increasing by 32.1%, 39.7%, and 31.1%, respectively. The proportion of fine roots was increased in ICNM, and the average diameter of roots decreased by 10.7% compared to TM. There was a significant negative correlation between mean root diameter and N uptake. In conclusion, the ICNM strategy was beneficial to form a good root system configuration, promote the development of shoot biological potential, increase tomato yield, maintain fruit quality, increase N uptake, and reduce environmental risks.

**Keywords:** greenhouse tomato; cultivation of east-west planting; slow and controlled-release fertilizer; root architecture

## 1. Introduction

Greenhouse vegetables are a vitally important industry in China with a share of 19.1% in total agricultural output [1]. The large-scale greenhouse facilities in China account for more than 85% of the world's greenhouse agricultural production area [2]. In the past several decades, the north-south planting management strategy has formed in the vegetable production in solar greenhouses in northern China (Figure 1a), based on large inputs of manual practices. The cultivation method of north-south planting is convenient for farmers to manage the cropland, but there are great limitations in developing mechanized production. For example, when the agricultural equipment is used to make plots in a solar greenhouse, because of the short distance between north and south directions (generally 6–10 m), the equipment needs to turn around repeatedly, which expends the cost of vast time and labor. Therefore, it is necessary to change the traditional management strategy of cultivation in solar greenhouses, to choose the east-west cultivation to facilitate mechanized operation (Figure 1b), and explore a new way of effective integration between agricultural machinery and agronomy [3]. At the same time, the utilization rate of water

and fertilizer in greenhouse vegetable production is low, which leads to waste of resources and environmental pollution, and some areas of greenhouse vegetable production have suffered from severe soil damage and nutrient loss [4,5], which is usually accompanied by the phenomena including diseases, even yield reduction and quality degradation of vegetable crops [6]. Under the condition of east-west cultivation, reducing fertilizer input is a necessary measure to avoid resource waste and improve fertilizer utilization efficiency. However, it is challenging for greenhouse vegetable production how to keep the stable yield or even increase the yield and improve nutrient utilization efficiency to guarantee coordination with environmental protection [7,8]. It is found that the integration of water and fertilizer is widely used in greenhouse vegetable production, and the production efficiency and modernization degree have been improved, and 75% of farmers in greenhouse vegetable production in Beijing used drip irrigation fertilization technology, but the irrigation regime still needs to be further optimized [9]. Besides, one-time basal application of slow/controlled release fertilizer and organic fertilizer is beneficial to reduce N loss in the agricultural system of high-yield tomato. Compared with conventional cultivation, the residual amount of nitrate-N in 0–100 cm soil profile is decreased by 21.0–59.8%, and then the leaching loss is reduced using slow release fertilizers [10].

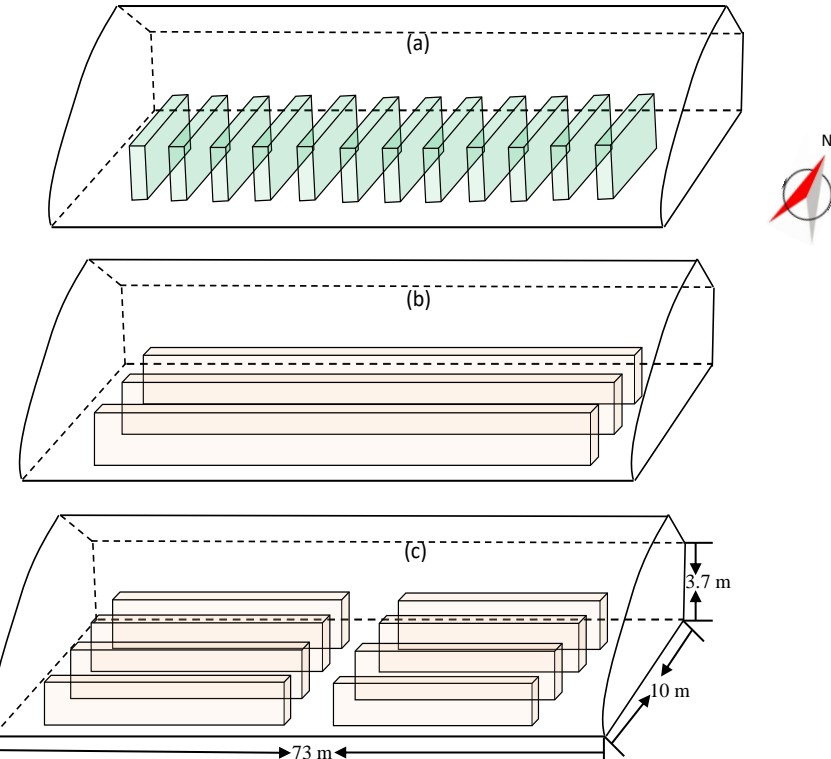

**Figure 1.** Comparison between traditional north-south cultivation (**a**) and east-west cultivation ((**b**,**c**) with a central belt) in solar greenhouses, Northern China.

On the premise of high yield, it is necessary to consider root distribution and architecture to promote nutrient efficiency, especially individual root architecture [11]. Of course, it was more important to further clarify the mechanism and regulation of efficient nutrient utilization in the root layer of high yield population based on individual root architecture. When there is a new competition for nutrients, the characteristics of root uptake would be changed obviously and adaptive plastic response would appear [12,13]. Therefore, it is more meaningful to study the root layer structure with a certain population density. Hodge et al. (2009) summarized that the hyperplasia of lateral root is beneficial for forming fine roots with strong absorption activity, and then improving the efficiency of root absorption, but it is worth pointing out that higher or lower one-side N supply would



inhibit the hyperplasia of lateral root [14]. The controlled release fertilizers could change the rule of N supply and greatly reduce the intensity of N supply in the early stage, but these fertilizers were always fixed at the initial position after being applied to soil, which was more conducive to the regulation of root architecture in terms of intensity and space of N supply. However, the single control technology could not coordinate quality and yield with high efficiency and environmental protection [15]. Previous studies have mostly focused on the optimization of a single technology or single factor, with a lack of consideration for comprehensive cultivation measures, and the efficiency of production has still been low. These multi-objectives, such as yield, quality improvement, less impacts on environment, and reduced labor input in vegetable cropping, are still hard to meet. On the basis of this agricultural problem, a field experiment was conducted to verify the hypothesis that altered cultural management, and a modified fertilizer delivery system could increase yield, improve root architecture, and reduce nitrate residual. In this experiment, a new integrated management strategy with the east-west cultivation cooperated with the application of new fertilizers was set up in a greenhouse. Our objectives are as follows: (1) to investigate the effects of the integrated crop-nitrogen management strategy (ICNM) on tomato yield, root architecture, soil residual N; and (2) to study the correlation of root architecture on N utilization factors.

## 2. Materials and Methods

### 2.1. Experimental Site

The field experiment was conducted in 2019 at the Agricultural Technology Development Center of Hancunhe Village, Fangshan District, Beijing, China (39°36′13″ N , 115°56′32″ E). The greenhouse is a three-faced brick wall structure, topped with polyethylene film, oriented east-west, with a ridge height of 3.7 m, a span of 10.0 m, and east-west length of 73.0 m (Figure 1c). Top 0–20 cm soil was measured following standard procedures before experiment start. The soil contained 10.1 g kg$^{-1}$ of organic matter content, 0.56 g kg$^{-1}$ of total N, 29.62 mg kg$^{-1}$ of Olsen-P, and 123 g kg$^{-1}$ of available K. The temperature and relative humidity of air during the experimental period in 2019 are shown in Figure 2.

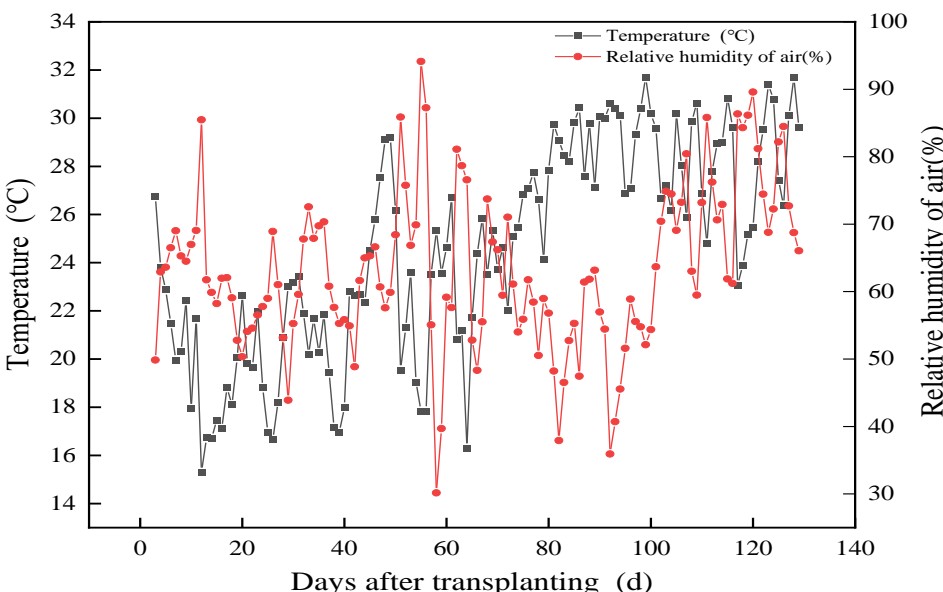

**Figure 2.** Daily mean temperature and relative humidity of air during the experimental period of tomato in 2019.

### 2.2. Treatments

Tomato (*Solanum lycopersicum* L.) variety Jinguan 58 was used. The greenhouse separated the eastern part and western part from the center line before the experiment (Figure 1c). The experiment used a split-plot design, two management strategies main-

tained in the main plots, and four fertilizer types allotted in the subplots. Each treatment combination was repeated three times with random distribution. The main plot treatments were the traditional management strategy of farmers located in the east area (TM, conventional thin planting: 33,000 plants ha$^{-1}$, four spikes, topdressing common solid water-soluble fertilizer with 200 kg N ha$^{-1}$), and the ICNM strategy was located in the west area (dwarfing dense planting: 66,000 plants ha$^{-1}$, three spikes, topdressing liquid water-soluble fertilizer with 200 kg N ha$^{-1}$). The subplot treatments were four different basal fertilizers, including CK (0 kg N ha$^{-1}$), BU (100 kg N ha$^{-1}$ of carbon-based urea), CU (100 kg N ha$^{-1}$ of controlled release urea), and CF (100 kg N ha$^{-1}$ of conventional compound fertilizer), respectively. The fertilizers and nutrient contents are shown in the Table 1.

**Table 1.** Nutrient contents of fertilizers used in the experiment.

| Fertilizers | Nutrient Contents |
| --- | --- |
| Urea ammonium nitrate solution (UAN) | N 32% |
| Ammonium polyphosphate (APP) | N 11%, $P_2O_5$ 37% |
| Potassium solution | $K_2O$ 30% |
| High N type water-soluble fertilizer | N:$P_2O_5$:$K_2O$ = 30–10–10% |
| High potassium type water-soluble fertilizer | N:$P_2O_5$:$K_2O$ = 15–5–30% |
| Controlled release urea (CU) | N 43% |
| Carbon-based N compound fertilizer (BU) | BU: N:$P_2O_5$:$K_2O$ = 14–14–14% |
| Conventional compound fertilizer (CF) | CF: N:$P_2O_5$:$K_2O$ = 17–17–17% |
| Calcium superphosphate | $P_2O_5$ 12% |
| Potassium sulfate | $K_2O$ 50% |
| N fertilizer inhibitor | Nitrification inhibitor dicyandiamide (DCD) and urease inhibitor N-butyl thiophosphoryl triamine (NBPT) |

*2.3. Fertilization and Crop Management*

The basal fertilizer was applied before plowing. The total amount of base fertilizer was 100 kg N ha$^{-1}$, 100 kg $P_2O_5$ ha$^{-1}$, and 100 kg $K_2O$ ha$^{-1}$, respectively. Topdressing was carried out for three times, including 57 days after planting (DAT, the first fruit-set stage), 67 DAT (fruit expending stage), and 79 DAT (peak fruiting stage). The drip irrigation system was used for topdressing. The main pipes were installed on the center belt between two main plots, with the same irrigation amount of 352 mm for two strategies. The solid water-soluble fertilizer included two products of high N type (30–10–10) and high potassium type (15–5–30), which were used in combination. The liquid fertilizer consisted of liquid N fertilizer (UAN), liquid phosphorus fertilizer (APP), and liquid potassium fertilizer. The total amount of topdressing was 200 kg N ha$^{-1}$, 60 kg $P_2O_5$ ha$^{-1}$, and 200 kg $K_2O$ ha$^{-1}$. Especially, CK treatment had no N in the base fertilizer and its top dressing was the same as other treatments. Ridge cultivation was conducted from east to west with a bed width of 1.0 m, height of 20 cm, and bed spacing of 40 cm. The density planting in the west area used a plant space of 20 cm, and the thin planting in the east area used a plant space of 40 cm on the bed.

*2.4. Sample Collection and Measurement*

The collection and determination of the soil sample: Soil samples were collected before transplanting, on the 20th DAT, 40th DAT, and after harvesting, respectively. At 20 DAT and 40 DAT, top 0–20 cm soils were collected, and afterwards, harvest soil samples were collected at the depths of 0–20 cm, 20–40 cm, 40–60 cm, 60–80 cm, and 80–100 cm. Three locations were taken from each experimental plot, and the soil samples were mixed uniformly and taken back to the laboratory. The fresh soil samples were used to determine inorganic N (ammonium-N and nitrate-N) and moisture content. The surface soil sample of 0–20 cm before transplanting was air-dried, and then the organic matter of soil (potassium dichromate-External heating method), total N (Kjeldahl method), Olsen phosphorus

($NaHCO_3$ extraction-Vanadium molybdenum blue colorimetric method), available potassium (ammonium acetate extraction-Flame photometry), pH (pH meter) (soil:water = 1:2.5, *w:w*), and EC (conductivity meter) (soil:water = 1:5, *w:w*) were determined and measured. The detailed procedures for the chemical analysis followed the methods described in a previous study of Bao [16].

Determination of fruit yield and quality: Ten continuous plants out of each plot were chosen as fixed yield collecting area. The amount of each pick of ripe fruits was recorded according to the planting density and different treatments. Three selected representative fruits from the 2nd to 3rd ear of mature tomato were taken back to the laboratory to determine the content of vitamin C (2,6-dichloroindophenol method), soluble sugar (Anthrone colorimetric method), titratable acid (Acid-base colorimetric method), lycopene (High performance liquid chromatography), and nitrate (Ultraviolet spectrophotometry) of which detailed procedures were referenced from the method of Bao [16].

Collection and determination of roots of tomato: At the late stage of the first ear enlargement (74 DAT), one plant's root was collected from each plot. Around the root, the three iron plates were inserted into designated positions in the left, right, and rear directions, respectively, and a square clod with a length, width, and height of 20 cm × 20 cm × 20 cm was taken out and placed on a 1 mm gauze net. The square clod was slowly rinsed with tap water to gradually remove the soil in order to obtain a complete root. Subsequently, the roots were placed in the sealed bag and marked and taken back to the laboratory for analysis. In the laboratory, the roots were rinsed with distilled water to remove impurities, then the roots were cut off one by one with scissors and put into the scanning disk to get root images by the Root Scanner (Epson, Suwa, Japan). Finally, the WinRHIZO software was used to analyze each image and obtain the length, diameter, surface area, and volume of the roots. The classification of length at different root diameters (with an increment of 0.5 mm from zero to the above 4.5 mm) was automatically shown in the results of the software. Here, we summed and classified it as 0 mm < D ≤ 0.5 mm, 0.5 mm < D ≤ 1.0 mm, 1.0 mm < D ≤ 2.0 mm, and D > 2.0 mm to evaluate the root architecture and used the fine root at 0 < D ≤ 0.5 mm and 0.5 < D ≤ 1.0 mm to find the relationship with soil nitrate-N.

*2.5. Data Analysis*

Data analysis was performed using the SPSS Statistics software 20.0. Under the same management strategy, one-way analyses of variance (ANOVA) followed by means comparison using Duncan's multiple range test at the level of 0.05 for tomato yields, quality, plant N uptake, root, and soil mineral N were performed. When comparing the interaction between two strategies, two-way analysis of variance (ANOVA) was performed with the general linear model procedure to calculate the effects of management strategies and fertilizer types on the investigated parameters. When the F value was significant, a multiple means comparison was performed with Duncan's test at the level of 0.05 and 0.01. Based on the data of root, SPSS 20.0 was used to analyze the correlation of all indicators, and the indicators that had significant correlation with root architecture were subjected to regression and linear or nonlinear model fitting. The correlation between the indicators was evaluated using the coefficient of correlations ($R^2$) and the root mean square error (RMSE). Moreover, the regression equations of the soil nitrate-N content, plant N uptake versus root length and diameter were computed. Linear fitting was well described by the relationships between fine roots at 0 < Diameter (D) ≤ 0.5 and residual nitrate-N, and between root average diameter and plant N uptake. The formulae for N uptake and partial productivity of N fertilizer are shown below.

$$\text{N uptake (kg ha}^{-1}) = \text{plant biomass (kg ha}^{-1}) \times \text{plant N content (\%)},$$

$$\text{Partial productivity of N fertilizer (PFP}_N, \text{ kg kg}^{-1}) = \text{fruit yield (kg ha}^{-1}) / \text{total N application rate (kg N ha}^{-1}).$$

## 3. Results and Analysis

### 3.1. Tomato Yield, Quality, and N Uptake under Different Management Strategies

There was no interaction between planting density and fertilizer on yield and absorption efficiency of tomato, but both factors-management strategies (M) and fertilizer types (F) had a significant effect on N partial productivity and yield of tomato (Table 2). Compared with the traditional management strategy (TM), the integrated management strategy (ICNM) achieved significantly higher N uptake, N partial productivity and yield of tomato, with an increase of 39.7%, 31.1%, and 32.1%, respectively. In TM, there were significant differences in N partial productivity and yield among different fertilizer treatments; the yield in CF treatment was significantly increased by 24.8% and 11.9% compared with that of CU and CK. For the CK treatment, with the lowest N input, the N partial productivity in CK was significantly higher than other treatments. In ICNM, there was a significant difference in yield of tomato among four fertilizer treatments. The yields in CU and CF treatment were significantly increased by 25.9% and 28.2%, respectively, compared with CK. At the same time, the individual plant's nutrient requirement decreased for more plants, and N supply of controlled release fertilizer could meet the growth of tomato and improve the yield of CU. In both TM and ICNM, the yield of tomato in CF was the highest.

**Table 2.** Effects of different treatments on plant N utilization and yield of tomato.

| | Treatments | Total N Content of Plant (%) | Total N Content of Fruit (%) | N Uptake $(kg\ ha^{-1})$ | PFP $_N$ $(kg\ kg^{-1})$ | Yield $(t\ ha^{-1})$ |
|---|---|---|---|---|---|---|
| Traditional management (TM) | CK | 1.58 ± 0.13 a | 1.97 ± 0.29 a | 124 ± 4.90 a | 373 ± 40.91 a | 74.6 ± 8.18 bc |
| | BU | 1.69 ± 0.20 a | 2.03 ± 0.12 a | 127 ± 19.08 a | 260 ± 46.19 b | 78.0 ± 13.86 ab |
| | CU | 1.67 ± 0.10 a | 1.86 ± 0.20 a | 115 ± 12.10 a | 223 ± 25.11 b | 66.9 ± 7.53 c |
| | CF | 1.81 ± 0.26 a | 2.16 ± 0.19 a | 138 ± 8.25 a | 278 ± 35.30 b | 83.5 ± 10.59 a |
| Integrated management (ICNM) | CK | 1.86 ± 0.23 a | 2.16 ± 0.22 a | 155 ± 26.48 a | 433 ± 79.70 a | 86.6 ± 7.34 b |
| | BU | 1.72 ± 0.20 a | 2.14 ± 0.12 a | 180 ± 36.31 a | 320 ± 28.96 a | 96.0 ± 8.69 ab |
| | CU | 1.92 ± 0.19 a | 2.19 ± 0.13 a | 189 ± 37.78 a | 362 ± 67.62 a | 109 ± 10.78 a |
| | CF | 1.72 ± 0.13 a | 2.16 ± 0.23 a | 181 ± 4.92 a | 369 ± 73.93 a | 111 ± 12.21 a |
| TM | | 1.69 ± 0.13 a | 2.00 ± 0.09 a | 126 ± 10.01 b | 283 ± 35.71 b | 75.7 ± 9.71 b |
| ICNM | | 1.80 ± 0.15 a | 2.16 ± 0.05 a | 176 ± 13.26 a | 371 ± 28.67 a | 100 ± 6.31 a |
| Significance levels | Management strategies (M) | ns | ns | ** | ** | ** |
| | Fertilizer type (F) | ns | ns | ns | ** | * |
| | M × F | ns | ns | ns | ns | ns |

Note: Different letters in the same column showed a significant difference, $p < 0.05$; * means that there is a significant difference at the level of 0.05, ** means that there is a significant difference at the level of 0.01, ns means that there is no significant difference.

As shown in Table 3, except for lycopene, there was no interaction between planting density and fertilizer on the quality of tomato, but the fertilizer had significant impacts on sugar and acid and Lycopene. Compared with TM, titratable acid content in ICNM was increased by 18.6%, and the sugar/acid ratio did not significantly decrease. In TM, the lycopene content in CF was the highest (3.26 mg 100 $g^{-1}$), which was significantly higher than the other three treatments, with an increase of 35.3–50.9%. In ICNM, there were no significant differences in nitrate content, Vitamin C, soluble sugar, and Lycopene among the four treatments. The titratable acid content in the CF treatment was significantly reduced by 21.4–38.1% compared to that of other treatments, and the sugar and acid ratio in CF was significantly improved by 33.4% and 28.8%, respectively, compared to the CK and BU treatments.

**Table 3.** Effects of different treatments on fruit quality of tomato.

| | Treatments | Nitrate (mg kg$^{-1}$) | Vitamin C (mg 100 g$^{-1}$) | Soluble Sugar (%) | Titratable Acid (%) | Sugar Acid Ratio | Lycopene (mg 100 g$^{-1}$) |
|---|---|---|---|---|---|---|---|
| TM | CK | 234 ± 43.4 a | 10.5 ± 1.09 a | 3.52 ± 0.57 ab | 0.39 ± 0.05 a | 9.12 ± 1.43 a | 2.16 ± 0.28 b |
| | BU | 231 ± 32.8 a | 11.4 ± 1.04 a | 3.38 ± 0.46 b | 0.45 ± 0.03 a | 7.52 ± 1.49 a | 2.39 ± 0.27 b |
| | CU | 232 ± 40.4 a | 11.0 ± 1.95 a | 4.67 ± 0.35 a | 0.47 ± 0.07 a | 10.14 ± 1.90 a | 2.41 ± 0.25 b |
| | CF | 239 ± 43.4 a | 11.2 ± 1.58 a | 3.64 ± 0.30 ab | 0.39 ± 0.06 a | 9.47 ± 2.17 a | 3.26 ± 0.39 a |
| ICNM | CK | 265 ± 57.5 a | 9.68 ± 1.15 a | 3.72 ± 0.31 a | 0.51 ± 0.04 a | 7.27 ± 0.38 b | 2.46 ± 0.17 a |
| | BU | 228 ± 28.8 a | 10.5 ± 0.39 a | 4.00 ± 0.36 a | 0.53 ± 0.07 a | 7.53 ± 0.35 b | 2.46 ± 0.34 a |
| | CU | 241 ± 13.5 a | 10.2 ± 1.06 a | 4.42 ± 0.49 a | 0.58 ± 0.08 a | 7.73 ± 1.47 ab | 2.22 ± 0.18 a |
| | CF | 208 ± 24.0 a | 10.9 ± 1.26 a | 4.07 ± 0.33 a | 0.42 ± 0.04 b | 9.70 ± 1.50 a | 2.38 ± 0.34 a |
| TM | | 234 ± 24.8 a | 11.0 ± 1.34 a | 3.81 ± 0.20 a | 0.43 ± 0.02 b | 8.94 ± 1.07 a | 2.56 ± 0.21 a |
| ICNM | | 236 ± 16.5 a | 10.3 ± 0.32 a | 4.05 ± 0.18 a | 0.51 ± 0.04 a | 8.00 ± 1.08 a | 2.38 ± 0.15 a |
| Significance levels | M | ns | ns | ns | ** | ns | ns |
| | F | ns | ns | * | * | * | * |
| | M × F | ns | ns | ns | ns | ns | * |

Note: Different letters in the same column showed a significant difference, $p < 0.05$; * means that there is a significant difference at the level of 0.05, ** means that there is a significant difference at the level of 0.01, ns means that there is no significant difference.

### 3.2. Tomato Root Architecture under Two Strategies

Except for the average diameter of root, the interaction between fertilization and planting density had a significant impact on root architecture. As shown in Table 4, the average root diameter in ICNM was significantly lowered by 11.9% compared to that of TM. In the TM system, there were significant differences in length of root and length density of root among treatments, and the length of root in CU and CF were 43.1–122% higher than CK and BU, respectively. In ICNM, there were significant differences in all indexes except the average diameter of root, and the lowest indexes occurred in the CU treatment, while the highest occurred in CK.

**Table 4.** Effects of different treatments on root architecture of tomato.

| | Treatments | Root Length (cm) | Root Length Density (cm cm$^{-3}$) | Surface Area (cm$^2$) | Average Diameter (mm) | Root Volume (cm$^3$) |
|---|---|---|---|---|---|---|
| TM | CK | 13,575 b | 1.70 b | 3530 a | 0.74 a | 174 a |
| | BU | 20,342 b | 2.14 ab | 23,922 a | 0.80 a | 98.3 a |
| | CU | 29,103 a | 6.07 a | 10,425 a | 0.74 a | 233 a |
| | CF | 30,144 a | 3.01 ab | 5590 a | 0.73 a | 132 a |
| ICNM | CK | 45,157 a | 5.64 a | 8905 a | 0.67 a | 227 a |
| | BU | 31,277 b | 3.91 b | 6686 b | 0.68 a | 202 a |
| | CU | 17,888 c | 2.24 c | 3514 c | 0.62 a | 71.6 b |
| | CF | 31,324 b | 3.92 b | 6921 b | 0.70 a | 188 a |
| TM | | 25,849 a | 3.23 a | 5133 a | 0.75 a | 172 a |
| ICNM | | 31,411 a | 3.93 a | 6506 a | 0.67 b | 159 a |
| Significance levels | M | ns | ns | ns | ** | ns |
| | F | * | * | ** | ns | * |
| | M × F | ** | ** | ** | ns | ** |

Note: Different letters in the same column showed a significant difference, $p < 0.05$; * means that there is a significant difference at the level of 0.05, ** means that there is a significant difference at the level of 0.01, ns means that there is no significant difference.

The roots were classified according to the different diameter ranges and expressed as root length (Figure 3). It could be observed that more roots were distributed within D < 0.5 mm, and the proportion of these fine roots was larger, accounting for 52.8% and 61.8% in TM and ICNM, respectively. In TM, there were significant differences in the three root classifications among different treatments, significantly more fine roots in CU than in other treatments, and the least of fine roots in CK. In ICNM, there were also significant differences in root classifications among different treatments, and the proportion of fine roots in CK was the largest, and the smallest was in CU. The fertilizer treatment has sharply different impacts on the classifications of fine roots both in TM and ICNM management. The proportion of fine roots with D < 0.5 mm in CU was the highest in TM, but the lowest

in ICNM. The proportions of roots at D < 0.5 mm in CK, BU, and CF were significantly increased in ICNM, indicating that applying different fertilizers after increasing the planting density could significantly change the distribution of root diameters and thus improve root N absorption.

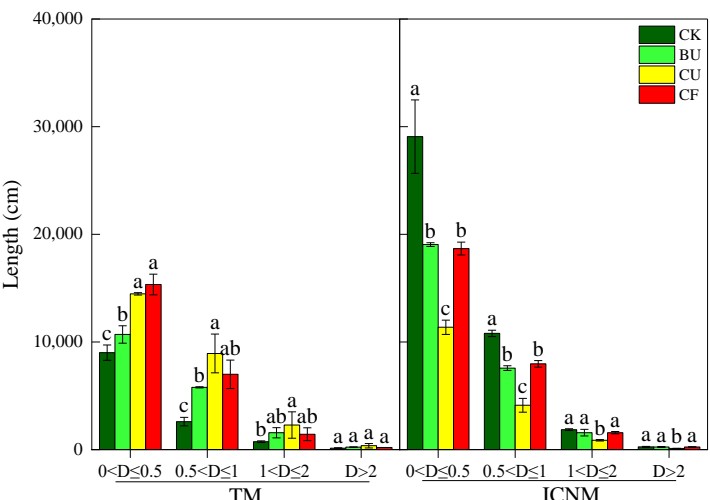

**Figure 3.** The length classification of root with different diameters. Different lowercase letters indicate significant differences at *p* = 0.05, the same below.

There were significant differences of root weight among treatments (Figure 4). Under TM strategy, the root fresh weight of CU was 28–44% higher than that of other treatments. The root dry weights of CU and CF were significantly higher than those of CK and BU, with increases of 46–56% and 53–64%. Under the ICNM strategy, however, the root fresh weight of CK and CF was significantly higher than that of BU and CU, with an increase of 21–69%, and the root dry weight of CU was 36–42% lower than that of other treatments.

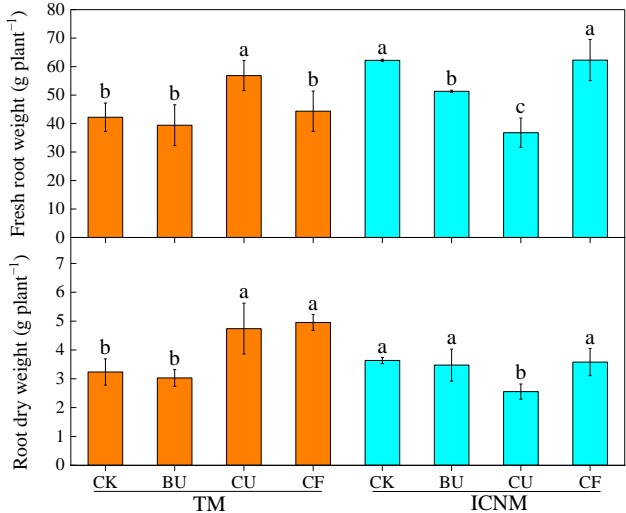

**Figure 4.** Dry and fresh root weight of different treatments. Different lowercase letters indicate significant differences at *p* = 0.05, the same below.

### 3.3. Soil Nitrate-N Residue after Harvest

Nitrate-N is relatively easy to move with the water in soil and easily leaches into the deep soil. As shown in Figure 5, the accumulation of nitrate-N in soil in TM was significantly higher than that in ICNM, especially in the 0–40 cm of soil profile, indicating that the absorption efficiency of root for nitrate-N was low in TM. The ICNM was conducive to reduce leaching of nitrate and improve absorption efficiency of root. In TM, there were

significant differences in different soil layers (0–20 cm, 20–40 cm, 40–60 cm, 60–80 cm, and 80–100 cm) among four treatments (CK, BU, CU, and CF), and CU and CF treatments lowered the risk of N leaching to the 60–100 cm of soil layer. In ICNM, the nitrate-N content in 0–60 cm of soil layer in CU was higher, which indicated that there could be a lag in N release; soil nitrate-N in CF treatment did not accumulate in the upper soil or the lower soil layer, indicating that the utilization efficiency of N was higher.

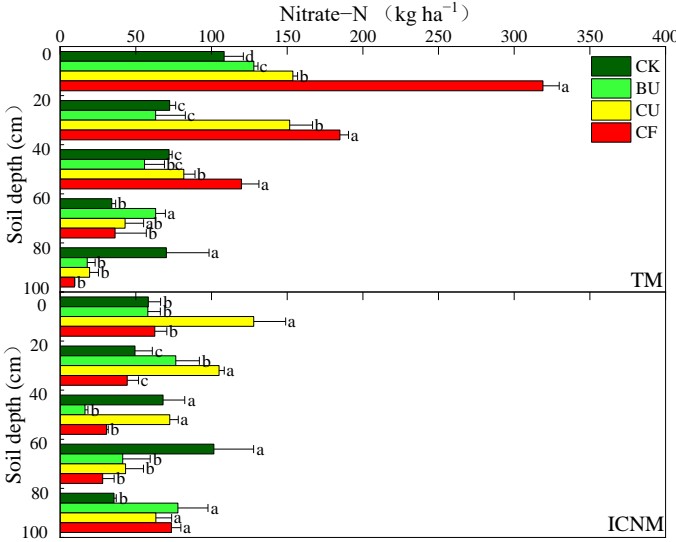

**Figure 5.** Distribution of nitrate−N in 0−100 cm of soil layer after harvest. Different lowercase letters indicate significant differences at *p* = 0.05.

### 3.4. Relationship between Plant Root Architecture and N Utilization

Considering the planting density and fertilizer comprehensively, it could be seen from Figure 6 that the fine roots with $0 < D \leq 0.5$ had a significant negative correlation with nitrate-N content. With the increase of fine roots, the content of nitrate-N accumulation in soil was decreased and the absorption and utilization efficiency of N fertilizer were enhanced. The value of $R^2$ was 0.4485, and the root mean square error (RMSE) was 9.33. There was a significant negative correlation between the average diameter of root and N uptake. With the decrease of the root diameter, an increase of N uptake was observed, that was, the fine roots is the core force responsible for absorbing nutrients from soil solution, with the $R^2$ of 0.3375 and RMSE of 14.47.

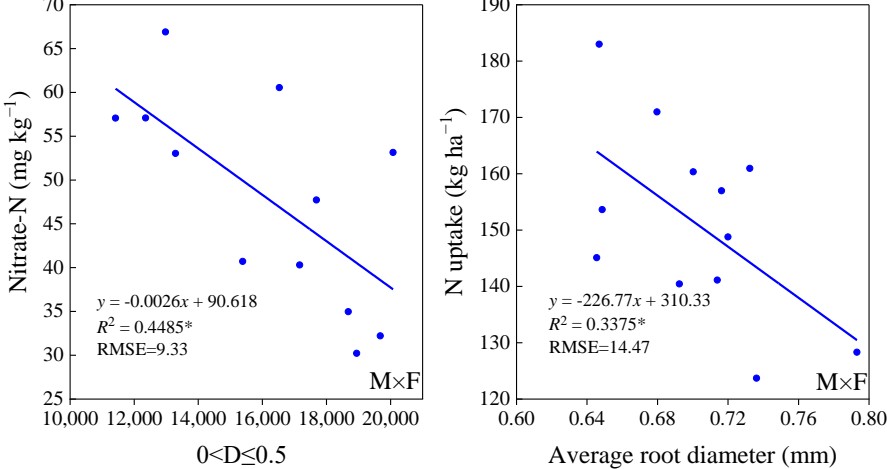

**Figure 6.** Relationship between fine root at of $0 < D \leq 0.5$, average root diameter of and nitrate-N content, N uptake. * means that there is a significant difference at the level of 0.05.

## 4. Discussion

### 4.1. Effects of Different Management Strategies on Root N Absorption of Tomato

Improved nutrient management plays an important role in increasing crop yield and N use efficiency [17]. Many previous studies have indicated that single nutrient management measures could only maintain vegetable yield. The development of novel, integrated strategies that combine soil, crop, nutrients, and water management will be necessary to sustainably increase vegetable yield and reduce environmental impacts, to meet the demand for vegetables by a growing population. A systematic and integrated approach has been proved to increase crop yield with lower environmental cost for global crop sustainability. In a plastic greenhouse of central China, the total fruit yields and N uptake of integrated soil-crop system management were 37.3% and 24.0% higher than that of the farmers' practice treatment [18]. The U-NSGA-III algorithm developed in the USA optimized a farm level agricultural production system against a myriad of soil, crop, and climate objectives, with reduction of water usage by 48%, N usage by 26%, and N leaching by 51% [19]. This study here showed that the yields and N uptake of ICNM were increased by 32.1% and 39.7% than that of the TM. Therefore, in this study, integrated crop-nitrogen management such as east-west planting system, close planting, and water and fertilizer integration suitable for mechanical operations were adopted to improve water and fertilizer utilization efficiency and reduce environmental pollution on the premise of stable and increased production.

Root morphology of plants determines its ability to absorbing water and nutrients from soil, and it was an important organ that affected the yield formation [20]. It was known that the status of root could be reflected through length, surface area, average diameter, and other indicators [21]. According to the past studies, forming an ideal or optimal root architecture was an important way to further improve nutrient utilization [22,23]. More than 90% of tomato roots were distributed in the top 0–40 cm of soil layer [24]. The results of this study showed that fertilizer treatments had significant effects on the root architecture of tomato. Some studies showed that the elongation of root was promoted by low concentrations of nitrate and inhibited by high concentrations [25]. Appropriate low N conditions could stimulate the growth of lateral roots [26]. At the same time, appropriate low N could increase the number of lateral roots, the total length of lateral roots and the length of lateral roots per unit axial root, but the density of root length was decreased [27,28], which was basically consistent with the results of this study. In ICNM, on the premise of equal supply of water, because the water consumption of dense planting system was larger, the soil water became less, and the early-stage N deficiency in CK and water deficiency in the surface layer together resulted in a less distribution to above ground and more roots in top soil [29]. While the early-stage N supply in CF was sufficient, the root architecture was coordinated with the shoot, therefore, the yield of tomato was higher. However, CU had no stimulating effect on the growth of root. The release of CU was slow, which was insufficient to support the nutrient demand of high density cultivation. However, in TM with thinning planting, the N supply from CU just could form the root architecture configuration to keep utilization efficiency of water and fertilizer. In addition, in ICNM, the growth space of the plants was relatively small, the roots could interact with each other, and the proportion of fine roots was increased, so the stimulation effect on fine roots in BU and CU treatments was not reflected. In TM, there was a large growth space for roots and little interaction between roots, and CU and CF had a positive stimulating effect on roots. In summary, the longest length of root could not be the best one, and an appropriate proportion of different diameters is necessary. At the same time, CU was mainly applied in the 0–10 cm of soil layer during fertilization, further adjustments were needed, such as reducing the release period and adjusting the location of fertilization, which could achieve the synergistic effect of yield and efficient nutrient absorption.

### 4.2. Synergistically Improve N Absorption and Utilization Ratio, Yield, and Quality of Tomato

To improve the utilization rate of fertilizer, scientific and reasonable control should be taken into consideration during the application of base fertilizer according to the soil background fertility, and reasonable topdressing should be carried out to match nutrient needs of tomato plants at growing and developing stage, thus resulting in improving yield and quality to some extent [30]. Previous results show that the average greenhouse tomato yield in China is 63–128 t ha$^{-1}$ [31–33]. In this study, the average tomato yield under TM strategy was 75.7 t ha$^{-1}$, and the average yield under ICNM strategy increased by 32% to 100 t ha$^{-1}$. The tomato yield under ICNM strategy is at the higher level in China.

In this study, the planting density and the new liquid fertilizer with urea inhibitors and nitrification inhibitors were used to comprehensively regulate the management strategy. Different basic fertilizers could shape better roots, and the new type of liquid fertilizer by topdressing was more conducive to improve root absorption and reducing nutrient residue. UAN was a liquid N fertilizer containing three N forms (nitrate-N, ammonium-N, and amide-N). The nitrate-N could be directly absorbed by the root architecture, while the ammonium-N and amide-N could be absorbed after nitration and hydrolysis, so the N supply in the time dimension was gentler. At the same time, due to the action of inhibitors, the existence time of ammonium-N was prolonged, which changed the ratio of ammonium and nitrate in soil and promoted the growth and nutrient absorption of root [34]. Therefore, the liquid fertilizer could not only provide nutrients quickly, but also slow down the conversion rate of ammonium-N to nitrate-N, thus reducing the N supply intensity of N fertilizer in a few days after fertilization. It was more consistent with the nutrient absorption characteristic of vegetables, and helped improving the N absorption efficiency and reducing soil N leaching. Compared with urea, the N content in UAN liquid fertilizer was 30.4% lower, and it was evenly mixed with the liquid inhibitor, which could avoid N loss and uneven mixing with low-dose inhibitors in urea form, thus improving the N utilization efficiency of UAN. At the same time, east-west cultivation was beneficial to lower the cost of drip irrigation facility and facilitate mechanized production, promoting the efficiency of drip fertigation and reduce the amount of N loss at early growth stage, further improving the water and fertilizer use efficiency, and reducing the labor input in the east-west vegetable cultivation [35,36]. Besides, various measures are taken to comprehensively control and explore the biological potential of root and shoot in order to increase yield production and efficiency and to reduce N pollution risk [37].

In ICNM, the quality of tomato did not decrease, and the partial productivity of N fertilizer and the yield of tomato were significantly increased by 31.1% and 32.1%, respectively. In terms of fertilizer types, CF treatment in two management strategies had a good effect on yield and quality of tomato, and the comprehensive control of CF in two strategies could improve the utilization rate of N fertilizer and the yield and quality of tomato. However, in ICNM, the indicators of root in CK were significantly higher than those in other treatments, but the yield of tomato was low, indicating that there was a risk of yield reduction.

### 4.3. Correlation between Root Architecture and Nutrient Absorption and Utilization Efficiency in Tomato

Root architecture is formed by the interaction of roots and environmental factors during the development process. The effect of nutrient supply on root development starts from the seedling stage. The change of root architecture is a complex process. Besides genes and soil environment, planting density and agronomic measures will also have impacts on root architecture. This study showed that both plant density and N fertilization had effects on root architecture. Dense planting increased root length by <0.5 mm (Figure 3) and decreased the average diameter of the root (Table 3), which provided support for synergistic nutrient absorption and the yield of tomato. At the same time, a large amount of nutrient loss in the vegetable field was directly related to the low absorption efficiency of root architecture of crop and unreasonable distribution. This study showed that fine

roots with $0 < D \leq 0.5$ had a significant negative correlation with nitrate N content. With the reduction of fine roots, the accumulation of nitrate-N in soil was increased, and the absorption and utilization efficiency of N fertilizer was decreased, which increased the risk of nitrate-N leaching into deep soil. There was a significant negative correlation between the average diameter of root and N uptake. With the decrease of the diameter, the fine roots enhanced nutrient uptake (N source) in soil which may be attributed to the resulting increased absorptive surface area of the root system for water and nutrients. However, further research will help to enhance the validity of the findings.

## 5. Conclusions

Different fertilizer treatments had a great influence on the architecture of root and the yield of tomato in TM, and a better management of ICNM could increase the yield of tomato by 32.1% than that in a traditional management of the TM strategy. With the significant root surface increase, the CF treatment could significantly increase the yield of tomato by 28.2%, keep the stable quality, and help to form better root architecture and promote N absorption in ICNM. There was a significant negative correlations between fine roots with $0 < D \leq 0.5$ and soil nitrate-N content, and between average diameter of root and plant N uptake. It can be summarized that increasing the number of fine roots can improve N absorption and utilization, and thus reduce nitrogen-related environmental risks.

**Author Contributions:** Conceptualization: J.Y., J.W. and G.Z.; methodology, C.L., Y.L. (Yahao Li), D.C. and J.Y.; data curation and formal analysis, C.L. and Y.L. (Yahao Li); original draft preparation, Y.L. (Yahao Li); writing, review and editing, C.L., J.Y., Y.L. (Yanmei Li), R.T., G.Z. and J.W. All authors have read and agreed to the published version of the manuscript.

**Funding:** The National Key Research and Development Projects: 2022YFE0199500; the National Key Research and Development Projects: 2017YFD0800405; Beijing Rural Revitalization science and technology project: 489300; Beijing Agricultural Technology Project: 2018021; Demonstration Program of Beijing Academy of Agriculture and Forestry Sciences: 20191201, 2019005.

**Institutional Review Board Statement:** Not applicable.

**Informed Consent Statement:** Not applicable.

**Data Availability Statement:** Not applicable.

**Acknowledgments:** The authors would appreciate Bingrui Lian and Zhiying Yang for their help in field works.

**Conflicts of Interest:** The authors declare no conflict of interest.

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
