# Peer review of "Integrated Crop-Nitrogen Management Improves Tomato Yield and Root Architecture and Minimizes Soil Residual N"

_agronomy, doi:10.3390/agronomy12071617_

Round 1
Reviewer 1 Report
Although the manuscript shows a lot of hard work by the authors, the manuscript is confusing in presentation of the procedures, data analysis and the results derivation.
Specific comments:
1. The manuscript lacks in the scientific presentation of the overall research.
2. The study is based on a single year. An additional year will be needed to verify that the results are consistent in at least two time points.
3. The research design is not clear in the manuscript. The plant population size is not clear in terms of data points as the ANOVA tables are not presented in the paper.
4. The authors described the greenhouse well but did not mention the air temperature and relative humidity maintained in the greenhouse throughout the crop cycle.
5. The yield of tomato seems very impressive but lacks discussion of comparison with past / recent yield by other researchers. What is the average tomato yield in general? Are the calculations done properly? Is this the record high in China and / or the world?
Reviewer 2 Report
This is a comprehensive manuscript talking about using ICNP to improve tomato yield and root architecture, very informative and has good logic and flow. Just few comments:
· Introduction: are there any differences between north-south and east-west planting in terms of water and nutrient use efficiency?
· Discussion: section 4.2, line 375-383, plant roots can also uptake ammonia N and will assimilate immediately, so it won’t reduce N supply intensity. I think instead of talking about supply intensity, you could discuss some differences in terms of nitrate N and ammonia N ratio in your products.
Reviewer 3 Report
The work is a response to increasing consumption and the consequent need to rationalize food production. The proposed modifications in tomato breeding significantly affect not only the increase of yield but also the decrease of labor input. The authors have developed an efficient and relatively simple experiment, the assumptions of which can certainly be implemented in tomato breeding on a larger, industrial scale. However, I have a few comments mainly on the description of the research methods and the statistical analyses used.
The abstract is too long and is practically a summary of the results. It mentions percentage changes of all tested parameters, as a result of which the reader is flooded with quantitative data from the very first lines. The abstract is supposed to briefly introduce the issue, describe the most important methods and the main results of the work and present the main conclusion. I suggest modifying this part of the manuscript.
The authors did not formulate a clear purpose for the paper. The end of the Introduction lists the expected effects of the experiment, but no objective. For this research, the objective was to verify the hypothesis that altered cultural management and a modified fertilizer delivery system could increase yield, improve root architecture, etc. It is important to remember that in a research study, you are TESTING the effect of a specific factor on the system under study, not PROVING the validity of a predetermined conclusion.
Chapter 2.2 'Experimental materials' is written in a very laconic style with the use of sentence equivalents and no verbs. I believe that it needs to be rewritten because in this form it does not meet the standards of scientific publications. What is more, the chapter lists ready-made basic fertilizers and the next chapter (2.3) describes fertilizer treatments whose composition is based on chemicals listed in chapter 2.2. I think that for better clarity and readability, the composition of fertilizer treatments should be described in a table. It would also be useful to separately collect and describe all abbreviations used in the text (treatments, DAT, etc.)
In Fig.1, the dimensions of the greenhouse and plant stands should be added. These values are given in the text, but would be easier to read in the figure.
Some parts of the Results belong in the Discussion: 'The yield reduction in CU...' lines 224-227; ‘So it could be summarized…’ lines 258-262.
The abbreviation 'Avgdiam' in Table 3 and Figure 5 should either be explained or replaced with the full name.
Figure 3 is poorly described, the description here only applies to CU treatment.
The caption for Figure 4 is not very informative.
Modification of root system architecture is discussed a lot in the paper. Of course, the authors give the relevant morphometric data, but, I think, showing photographs of roots growing in different treatments would allow the reader to simply see these modifications.
I have some objection to the statistical analysis. The authors indicated that they used Duncan's test to compare the features under study. This is a post hoc test, which according to statistical methods is performed after the main test that examines differences between means, so after the ANOVA test. However, the description of the statistical analysis did not mention the use of ANOVA. This is all the more puzzling since Tables 1, 2, and 3 describe the interaction (or lack thereof) between two factors (M and F) that may have influenced the means of the traits studied. Such interactions are examined with a two-factor analysis of variance test!
Moreover, the results given in Tab. 1, 2, 3 prove the differences between the means of some traits in different treatments , for example, Tab. 1 last column, yield in CK, BU, CU, CF differ as given: bc - ab - c - a. This means that when comparing management strategies we cannot treat this trait as uniform and compare the mean of all treatments from TM variant with the mean of all treatments of ICMN variant. In this situation we compare the mean of yield in CK and TM with the mean in CK and ICMN, so we have a comparison of two means and consequently, the Student's t test or its non-parametric equivalent should be used. The manuscript does not mention the use of such tests.
Figure 2 presents the length classification of root with different diameters. I understand that the WinRhizo software automatically creates root classes due to the root diameter, but this classification and the reason for its use should be mentioned in the methods.
I also want to point out the authors' firm conclusion regarding the correlation between root diameter and N uptake (Fig. 5). Of course, statistical analysis showed a significant correlation there, but it is a rather weak correlation (R^2=34%). Note that removing the three or even just the two extreme points on the graph would leave a point cloud occupying a circular area, indicating a complete lack of correlation. This result should be discussed in the Discussion.
The Discussion completely lacks direct references to the authors' own results (figures, tables, etc.). I think this should be supplemented.
Line 345: There is a reference to other researchers’ studies on root hair. I do not understand this reference since the authors of the manuscript did not study the effects of the treatments used in their experiment on root hair.
In the Conclusions, the authors state: 'Different fertilizer treatments had a great influence on the growth of root...'. However, such a conclusion is not justified based on their study because the paper did not examine root growth in the strict sense of the term.
The text occasionally contains typos (for example, line 72 'increasing the yield' instead of increase the yield; line 90 'Summarized' - unnecessary capital letter) or minor stylistic errors (mostly in Methods). This does not affect the quality of the work, but it is worth reviewing the text for these before final submission for publication.
I didn't check references.
Round 2
Reviewer 1 Report
Comments and suggested corrections/updates
Comments
Comment 1: The authors have revised and improved the manusript.
Comment 2: ANOVA tables were not provided by the authors. In ANOVA tables, I was looking for tables with column titles such as “Source of Variation”, “Degrees of Freedom(DF)”, “Sum of Squares(SS)”, “Mean Sum of Squares(MSS)”, and “Variance Ratio (F)”. Since many other published papers are also seen without actual ANOVA tables, I think the tables presented by authors from the results output of ANOVA can also be accepted. The authors have described ANOVA procedure well in the texts of the manuscript.
Suggested corrections/updates
Line 65: Please replace “has” with “have”.
Line 80(a): Please replace “to form” with “for forming”.
Line 80(b): Please replace “improves” with “impoving”
Line 81: Please replace “pointing” with “pointing out”.
Line 83: Please replace “reduced” with “reduce”.
Lines 88 through 90: Please replace “Previous studies mostly focus on the optimization of single technology or single factor, which is lack of considering comprehensive cultivation measures, and the efficiency of production is still low” with “Previous studies have mostly focused on the optimization of a single technology or single factor, which was lack of consideration for comprehensive cultivation measures, and the efficiency of production has still been low”.
Line 90: Please replace “quality improving” with “quality improvement”.
Line 99: Please replace “to qualify the effects of root architecture on the correlations between N utilization factors” with “to study the correlation of root architecture on N utilization factors”.
Lines 110 through 113: Please replace “air humidity” with “relative humidity of air”. Please also do the same in the Figure 2.
Line 114: Please replace “over the period 2019” with “in 2019”.
Line 116: Please replace “variety of Jinguan 58” with “variety Jinguan 58”.
Lines 128 and 129: Please replace “The abbreviations and details of all the fertilizers and 128 nitrogen inhibitor were showed in the table 1” with “The fertilizers and nutrient contents have been shown in the Table 1”.
Line 130: Please replace Table 1 title “Abbreviations and compositions of fertilizers used the experiment” with “Nutrient contents of fertilizers used in the experiment”.
Line 114: Please replace the Table 1 contents with the following:
Table 1. Nutrient contents of fertilizers used in the experiment
Fertilizers |
Nutrient Contents |
Urea ammonium nitrate solution (UAN) |
N 32% |
Ammonium polyphosphate (APP) |
N 11%; P2O5 37% |
Potassium solution |
K2O 30% |
High N type water-soluble fertilizer |
N:P2O5:K2O = 30-10-10% |
High potassium type water-soluble fertilizer |
N:P2O5:K2O = 15-5-30% |
Controlled release urea (CU) |
N 43% |
Carbon-based N compound fertilizer (BU) |
N:P2O5:K2O = 14-14-14% |
Conventional compound fertilizer (CF) |
N:P2O5:K2O = 17-17-17% |
Calcium superphosphate |
P2O5 12% |
Potassium sulfate |
K2O 50% |
N fertilizer inhibitor |
Nitrification inhibitor dicyandiamide (DCD) and urease inhibitor N-butyl thiophosphoryl triamine (NBPT) |
Line 156: Please replace “determined” with “used to determine”.
Line 157: Please replace “ammonium-n” with “ammonium-N”.
Line 166: Please replace “chose” with “chosen”.
Line 172: Please replace “referenced” with “were referenced”.
Line 206: Please replace “described” with “described by”.
Line 207: Please add a sentence “The formulae for N uptake and partial productivity of N fertilizer are shown below”.
Line 214: Please replace “the factor alone” with “both factors – management strategies (M) and fertilizer types (F)”.
Line 218: Please replace “treatment in” with “in”.
Line 219: Please replace “the different treatments” with “different fertilizer treatments”.
Line 223: Please replace “among four treatments” with “among four fertilizer treatments”.
Lines 228 and 229: In Table 2, please replace “ANOVA” with “Significance levels” on the first column towards the last section of the table.
Lines 228 and 229: In Table 3, please replace “ANOVA” with “Significance levels”.
Line 252: In the Table 4, please replace “ANOVA” with “Significance levels”.
Line 254: Please remove comma after length or replace “length,” with “length”.
Lines 260 and 261: Please replace “Fertilization treatments” with “The fertilizer treatment”.
Line 261: Please replace “fine root” with “fine roots”.:
Line 311: Please replace “environment” with “environmental”.
Line 328: Please replace “research form” with “past studies”.
Line 352: Please replace “could not the best” with “could not be the best”.
Line 353: Please replace “diameter” with “diameters”.
Line 386: Please replace “facilitate to” with “facilitate”.
Line 402: Please replace “Root architecture is formed by the interaction of root architecture and” with “Root architecture is formed by the interaction of roots and”.
Line 404: Please replace “has been started” with “starts”.
Line 406: Please replace “an impact” with “impacts”
Line 417 and 418: Please replace “With the derease of the diameter, the fine roots mainly absorved nutrients (N source) in soil” with “With the derease of the diameter, the fine roots enhanced nutrient uptake (N source) in soil which may be attributed to the resulting increased absorptive suface area of the root system for water and nutrients”.
Line 418 through 420: Please replace “However, though the statistical analysis showed a significant correlation between the average diameter and N uptakes, the correlation coefficient (R2=0.3375) was relatively low. More date and test are needed in future to consolidate the relationship” with “However, further research will help to enhance the validity of the findings”.
Line 424: Please replace “traditional mangement” with “a traditional management”.
Line 427: Please replace “correlation” with “correlations”.
Line 428 and 429: Pleae replace “there was a significant negative correlation between” with “between”.
Line 431: Please replace “reducing” with “reduce”.
